# Metabolomics-Guided Hypothesis Generation for Mechanisms of Intestinal Protection by Live Biotherapeutic Products

**DOI:** 10.3390/biom11050738

**Published:** 2021-05-15

**Authors:** Jiayu Ye, Lauren A. E. Erland, Sandeep K. Gill, Stephanie L. Bishop, Andrea Verdugo-Meza, Susan J. Murch, Deanna L. Gibson

**Affiliations:** 1Department of Biology, University of British Columbia, Syilx Okanagan Nation Territory, Kelowna, BC V1V1V7, Canada; jiayu.ye29@gmail.com (J.Y.); sand.gill01@gmail.com (S.K.G.); andrea.verdugomeza@ubc.ca (A.V.-M.); 2Department of Chemistry, University of British Columbia, Syilx Okanagan Nation Territory, Kelowna, BC V1V1V7, Canada; lauren.erland@ubc.ca (L.A.E.E.); stephanie.bishop@ucalgary.ca (S.L.B.); 3Department of Medicine, University of British Columbia, Syilx Okanagan Nation Territory, Kelowna, BC V1V1V7, Canada

**Keywords:** live biotherapeutics, gut–liver–brain axis, inflammatory bowel disease, immunometabolism

## Abstract

The use of live biotherapeutic products (LBPs), including single strains of beneficial probiotic bacteria or consortiums, is gaining traction as a viable option to treat inflammatory-mediated diseases like inflammatory bowel disease (IBD). However, LBPs’ persistence in the intestine is heterogeneous since many beneficial bacteria lack mechanisms to tolerate the inflammation and the oxidative stress associated with IBD. We rationalized that optimizing LBPs with enhanced colonization and persistence in the inflamed intestine would help beneficial bacteria increase their bioavailability and sustain their beneficial responses. Our lab developed two bioengineered LBPs (SBT001/BioPersist and SBT002/BioColoniz) modified to enhance colonization or persistence in the inflamed intestine. In this study, we examined colon-derived metabolites via ultra-high performance liquid chromatography-mass spectrometry in colitic mice treated with either BioPersist or BioColoniz as compared to their unmodified parent strains (*Escherichia coli* Nissle 1917 [EcN] and *Lactobacillus reuteri*, respectively) or to each other. BioPersist administration resulted in lowered concentrations of inflammatory prostaglandins, decreased stress hormones such as adrenaline and corticosterone, increased serotonin, and decreased bile acid in comparison to EcN. In comparison to BioColoniz, BioPersist increased serotonin and antioxidant production, limited bile acid accumulation, and enhanced tissue restoration via activated purine and pyrimidine metabolism. These data generated several novel hypotheses for the beneficial roles that LBPs may play during colitis.

## 1. Introduction

The use of probiotics to induce health benefits has existed for decades, albeit with conflicting outcomes [1]. This is partly because probiotics colonize, persist, and induce responses in the host in highly individualized patterns, limiting their universality [2]. The persistence of probiotics is heterogeneous [3], with low persistence rates in humans [4]. While not yet known, the native microbiome’s permissiveness to new species likely plays a role. In cases where health benefits are observed, commensal bacteria promote intestinal health through multiple mechanisms, including modulation of inflammatory cytokines, strengthening of the intestinal barrier, and normalization of dysbiosis [5]. Unlike conventional drugs that often target one pathway or effector, the advantages of using “bugs as drugs” are that bacteria can elicit multiple beneficial effects via various pathways. This multi-targeted approach may be more desirable in biologically complex conditions, such as inflammatory bowel disease (IBD).

Inflammation during IBD leads to oxidative stress in the intestine, which hinders effective persistence by commensal bacteria [6]. Indeed, the intestine is hostile to many commensal microbes during chronic inflammation, such as during an active IBD flare [7]. Therapies, like live biotherapeutics (LBPs), elicit health benefits via interaction with the gut microbiota and epithelium, influencing intestinal barrier function, the mucosal immune system, and the functional metabolic responses of the gut microbiome [8,9]. Live microbes, unlike molecules, have more potential for sustained protection if the microbe can persist in the inflamed intestinal niche.

Bio-engineering of probiotics such as *Lactococcus lactis* and *Lactobacillus casei* BL23 [10] to locally deliver therapeutic agents such as IL-10 [11], trefoil factors [12], and TNF-*α* neutralizing antibodies [13], has shown some benefit during gut inflammation. Similarly, *Bifidobacterium longum* bio-engineered to reduce reactive oxygen species by overexpressing manganese superoxide dismutase has demonstrated benefits in preclinical studies [14]. The FDA has approved clinical trials of bio-engineered LBP, SYNB1618, to treat the metabolic disease phenylketonuria (Synlogic, Cambridge, MA, USA) [15], indicating regulatory bodies will consider genetically modified organisms as viable therapeutics.

From several commercially available probiotics strains, *Escherichia coli* Nissle 1917 (EcN) stands out as an effective treatment against infectious [16] and ulcerative colitis [17]. EcN inhibits pathogenic bacteria colonization by secreting antimicrobial peptides [18], outcompeting pathogens for nutrient uptake [19], and displaying anti-inflammatory effects with reduced expression of colonic COX-2 (cyclooxygenase-2) and IFN-*γ* (interferon-*γ*) [20,21]. Therefore, engineering EcN may provide increased bioavailability of these bacteria to enable their beneficial effects in the intestine. Indeed, EcN overexpressing cystatin is immunomodulatory when compared to the unmodified strain [22]. Another probiotic that has previously shown protection against colitis is *Lactobacillus reuteri* [23]. In particular, it inhibits inflammatory cell infiltration [24] and decreases the expression of inflammatory markers such as myeloperoxidase and IL-6 [25].

Despite the promising use of these probiotics, a meta-analysis for EcN in IBD concluded that while EcN is equivalent to mesalazine in preventing disease relapse, its use in inducing the remission cannot be recommended [26]. This could be due to the lack of consistent persistence of the probiotic in the IBD intestine [4]. Hence, we hypothesized that optimizing LBPs with enhanced persistence in the inflamed intestine would help certain bacteria sustain their beneficial action in the intestine. To this end, our lab developed a bio-engineered LBP called SBT001/BioPersist, which was modified to enhance the parental probiotic EcN persistence in the inflamed IBD intestine, and SBT002/BioColoniz, which was modified to enhance the parental probiotic *L. reuteri*’s colonization in the intestine (PCT/CA2018/050188) [27]. The objective of the current study was to identify changes in the colon-derived metabolites during colitis in response to BioColoniz and BioPersist. We used an untargeted metabolomics approach to detect and identify metabolites in colitic mice colons treated with the LBPs or their unmodified parental strains. Our data analysis provides insights into how LBPs modulate specific metabolic pathways that impact both microbiome health and diversity as well as intestinal immunity. These data support several novel hypotheses on the mode of action and impact of BioPersist in the intestine.

## 2. Materials and Methods

### 2.1. Bacterial Culture

EcN and BioPersist were grown in Luria–Bertani overnight (LB, 10 g tryptone, 5 g yeast extract, and 10 g sodium chloride dissolved in 1 L of water, adjusting pH to 7.5) for 16 h at 37 °C under 180 rpm agitation. *L. reuteri* and BioColoniz were grown in De Man, Rogosa, and Sharpe broth (MRS, 10 g peptone, 10 g beef extract, 5 g yeast extract, 20 g dextrose, 5 g sodium acetate, 1 g polysorbate 80, 2 g dipotassium phosphate, 2 g ammonium citrate, 0.1 g magnesium sulfate, and 0.05 g manganese sulfate dissolved in 1 L of water, adjusting pH to 6.5) for 24 h at 37 °C static under anaerobic conditions given by the BD GasPak system (BD Biosciences, Franklin Lakes, NJ, USA). The mice were gavaged immediately with the fresh probiotics kept at room temperature. Parallel and to confirm the dose given to mice, 0.1 mL of the probiotics were plated on 100 × 15 mm agar plates of the corresponding medium and grown for 24 h at 37 °C for EcN and BioPersist, and for 48 h at 37 °C under anaerobic conditions for *L. reuteri* and BioColoniz.

### 2.2. Animal Experiments

The animal experimental protocol (A19-0286) was approved by the University of British Columbia’s Animal Care Committee in accordance with guidelines drafted by the Canadian Council and Animal care on the Use of Laboratory Animals. Female C57BL/6 mice (Charles River, Massachusetts, USA) were maintained under specific pathogen-free conditions, with a controlled temperature at 22 ± 2 °C, 12 h light/dark cycles, and autoclaved sterile drinking water and food (Teklad 7001, Envigo, Indianapolis, IN, USA) provided ad libitum. Mice were acclimatized and divided into four groups with *n* = 4 each, where each group came from various cages to minimize cage effects: *L. reuteri* DSM20016, BioColoniz, EcN (Mutaflor, Ardeypharm GmbH, Herdecke, Nordrhein-Westfalen, Germany), and BioPersist (16 mice total). Mice in the *L. reuteri* and BioColoniz received one single oral gavage dose of 0.1 mL of 2 × 10^9^ CFU/mL of the assigned probiotic. Mice in the EcN and BioPersist groups received 0.1 mL of 3 × 10^12^ CFU/mL of the assigned probiotic for three consecutive days via oral gavage. After that, mice were exposed to 3.5% dextran sodium sulphate (DSS, MP Biomedicals) via drinking water for 7 days to induce colitis. Then, mice were anesthetized with isoflurane and euthanized by CO_2_ asphyxiation, followed by cervical dislocation. Colon samples were collected, and flash frozen in liquid nitrogen. Samples were storage at −80 °C until further analysis.

### 2.3. Sample Preparation for Metabolomics Analysis

We extracted metabolites from the colon tissues based on previous methods [28,29,30,31]. Briefly, colon tissues were accurately weighed by difference using an analytical balance (Ohaus; VWR, Mississauga, ON, Canada) into a 1.5 mL microcentrifuge tube (Eppendorf, Mississauga, ON, Canada). Tissues were homogenized in 70% ethanol (200 µL 70% ethanol/100 mg tissue) with a disposable tissue grinder (Kontes Pellet Pestle; Fisher Scientific, Ottawa, ON, Canada). The resulting suspension was centrifuged at 3000× *g* for 3 min (Galaxy 16DH centrifuge, VWR, Mississauga, Ontario, Canada), and the supernatant was decanted and centrifuge-filtered using a 0.2 µm Ultrafree-MC centrifugal filter (Millipore-Sigma, Oakville, ON, Canada) at 3000× *g* for 3 min (Galaxy 16DH centrifuge). One hundred µL of the filtrate was transferred to an autosampler vial (300 µL polypropylene with pre-slit Teflon-coated caps; Waters Corp., Mississauga, ON, Canada) fitted with a conical bottom spring insert (250 µL glass; Canadian Life Science, Peterborough, ON, Canada) for ultra-high performance liquid chromatography-mass spectrometry (UHPLC-MS) analysis.

### 2.4. Metabolomics Analysis by UHPLC-MS

All samples were analyzed by UHPLC-MS with a Thermo Scientific Vanquish UHPLC binary system coupled to a Q-Exactive Hybrid Quadrupole-Orbitrap with a heated electrospray ionization (HESI) probe (Waltham, MA, USA). Analyte separation was achieved using a Phenomenex Kinetex EVO C18 column (2.1 × 150 mm, 1.7 µm; Torrance, CA, USA). A gradient-elution of 0.1% aqueous formic acid (Solvent A) and acetonitrile (Fisher, Ottawa, ON, Canada, Optima LC/MS grade; Solvent B) with initial conditions of 95:5, reaching 5:95 at 25.00 min and 95:5 at 30.00 min, was used; it had a re-equilibration time of 5.00 min to reach initial starting conditions and a curve 6 during the entire run. The column temperature was set to 30 °C, and the injection volume was 10 µL. For MS analysis, the scan range was set to full MS—we selected ion monitoring (SIM) with a scan range between 100–1200 *m/z*, a resolution of 70,000, a maximum injection time of 200 ms, and no fragmentation. The HESI source was run in positive ionization mode with a sheath gas flow rate of 65 units, an auxiliary gas flow rate of 20 units, and a sweep gas flow rate of 4 units. The spray voltage was 3.50 kV, while the capillary temperature was 300 °C, the auxiliary gas heater temperature was 500 °C, and the S-lens RF level was 50.0. Quality control (QC) samples were included as solvent blanks and alignment standards were analyzed in randomized order throughout the sample sets.

### 2.5. Data Processing & Annotation

Data including both samples and quality control samples were preprocessed with the online XCMS (https://xcmsonline.scripps.edu/) platform for peak detection, grouping, and alignment [32]. The key parameters were selected as: positive mode, 15 ppm (feature detection), mzdiff = 0.01, retention time correction method = obiwarp, and mouse database, with all other parameters set as default. The data matrix containing variables (retention time, mass, peak intensity) was obtained for further analyses.

### 2.6. Data Quality Analyses

Data quality and false discovery rates were determined by directional fold changes, differential volcano analyses, *t*-test, significant analysis of microarray (SAM), and empirical Bayesian analysis of microarray (EBAM) data using MetaboAnalyst Ver. 5.0 [33]. Colonic samples from four different LBP-treated C57BL/6 mice were fingerprinted, and the obtained data underwent a filtering and quality check procedure to produce four data sets with 4 biological replicates each (16 samples total). Next, a non-metric multidimensional scaling plot was performed to check the classification of the QC samples. Clustering of QCs (Appendix A) indicated the system’s stability and performance during analyses. Three pairwise comparisons were completed to investigate if enhanced colonization and persistence of probiotics possibly improved the DSS-caused inflammation. The three comparisons were: EcN and BioPersist, *L. reuteri* and BioColoniz, and BioPersist and BioColoniz.

### 2.7. Data Visualization

Unsupervised principal component analysis (PCA) and supervised partial least squares discriminant analysis (PLS-DA) were used to visualize data variances by MetaboAnalyst Version 5.0 [33]. Cross-validation of the PLS-DA was also performed to exhibit the classification performance with the top three components. The scores were plotted for downstream data exploration of each pairwise comparison. Statistical differences between groups were determined by the Wilcoxon rank-sum test for each metabolite in the MetaboAnalyst Version 5.0 platform.

### 2.8. Enrichment Analysis

The resulting data set of significant features was annotated using XCMS and exported. The monoisotopic mass, or the most simple isotopic mass of each annotated metabolite, was mined within our dataset for the exploration of pathway enrichment and metabolite identification with MetaboAnalyst ver 5.0 [33].

### 2.9. Pairwise Comparison

Three pairwise comparisons were applied to explore the signature pathways and metabolites: EcN vs. BioPersist, *L. reuteri* vs. BioColoniz, and BioPersist vs. BioColoniz. For all three pairwise comparisons, data were normalized to the former group. The data matrix of two groups exported from the XCMS online platform was loaded into the MetaboAnalyst platform (v5.0) [33] for further analysis. Of the three pairwise comparisons, *L. reuteri* vs. BioColoniz did not show good separation with the PCA plot. To eliminate the overfitting issue, which might have been raised by the PLS-DA, k-means clustering was performed.

### 2.10. Functional Analysis

Positive ion mode was applied for metabolite identification, with all other parameters set as default. The data were normalized, log-transformed, and scaled by autoscaling (mean-centering/standard deviation). Mouse BioCyc and mouse Kyoto Encyclopedia of Genes and Genomes (KEGG) databases were selected separately for further analyses. Monoisotopic mass in these two databases was used for metabolite matching. The data frame was exported and loaded to R studio for further data filtering. The simplest adduct of each metabolite was filtered out and retained with the dplyr package [34]. The output peak intensity tables were used for further pathway, network, and statistical analyses. The metabolites received from KEGG and BioCyc pathways displayed discriminate-annotated IDs. To compare the common metabolites received from the two databases, dplyr packages were used to apply feature-matching referencing both retention time and compound mass. because of the discrepancy of ID names between BioCyc IDs and the common chemical names, the KEGG ID was applied to the further analysis.

### 2.11. Pathway Analysis

The peak intensity table with KEGG IDs of each pairwise comparison was uploaded to the pathway analysis module of the MetaboAnalyst platform (v5.0). Compound labels were matched with uploaded KEGG IDs, and the *Mus musculus* KEGG database was selected for pathway analysis. Next, pathway enrichment analysis was performed with the global-test method to identify the most relevant pathways of the two groups. To define the connections between pathways, betweenness centrality was selected for topology analysis.

### 2.12. Biomarker Analysis

Biomarker analysis was additionally performed in the MetaboAnalyst platform to identify key metabolites that may be driving differences between the treatment groups. Data were processed using the support vector machine (SVM) algorithm. The receiver operating characteristic curve (ROC) analysis was used to evaluate the performance of the biomarker model using default settings and fitting to a 15-factor model.

### 2.13. Correlation-Based Network

Pairwise peak intensity tables were subjected to the Correction Calculator (v1.0.1) [35] of the metabolomics analysis. The data were normalized using the “log-2 transform data” and “autoscale data” options. Next, the annotated data were imported to Metscape version 3.1 in Cytoscape version 3.8.2 for visualization [36]. Correlation networks were calculated, tested, and structured by applying Pearson’s correlation coefficient (Pearson’s R). The threshold of correlation parameter was set as −1 to −0.5 for negative correlation and 0.5 to 1 for positive correlation. Partial correlations were calculated with the debiased sparse partial correlation (DSPC) algorithm [35]. The significant network correlation (*p* < 0.05) was obtained by setting the threshold *p*-value as 0.05.

### 2.14. Pathway-Based Network

Features annotated by the MetaboAnalyst platform (v5.0) with peak intensity were uploaded to Metscape version 3.1 in Cytoscape version 3.8.2 directly for building pathway-based networks. Data were matched to the mouse KEGG database and important individual pathways were selected to build sub-networks to better visualize the important features. Each sub-network was built by pathway names, which only included the metabolites of the selected pathway. The *p*-value was set as 0.05 with the fold-change as 1.5.

## 3. Results

### 3.1. The Comparison between EcN and BioPersist

#### 3.1.1. The Administration of EcN and BioPersist Shows Discriminate Features from DSS-Treated Mice Colons

A total of 94,235 features were called from the UHPLC-MS profile, and matching to the BioCyc database yielded 686 known metabolites, whereas annotation using the KEGG database yielded 839 unique metabolites (Table 1). All data were normalized to the EcN group. There were 533 common features the KEGG database shared with the matching compounds from the Biocyc database. Similarly, 483 metabolites from the Biocyc database were found in the matching compound list referring to the KEGG database (Table 1). The PCA (Figure 1A) and PLS-DA (Figure 1B) analyses with Components 2 and 3 both accounted for a significant portion of the variability in the first component. The PCA principal component one (PC1) accounted for 48.3% of the variability in the dataset in the unsupervised method, whereas PC2 and PC3 explained only a further 13.8% and 10.3%, respectively. The PLS-DA and Component 1 explained 47.2% of the variability, whereas Component 2 explained 9.1% and Component 3 explained 8%. Cross validation scores for the PLS-DA showed good model performances with R^2^ and Q^2^ of 0.99917 and 0.91013, respectively. Both methods showed no overlap in the 95% confidence intervals. This indicated that the administration of EcN and BioPersist affected the colonic metabolites in distinct patterns. Next, we explored the functional analysis with the mouse KEGG (Figure 1C) and mouse BioCyc (Figure 1D) databases. The common significant pathways with the KEGG and BioCyc databases were bile acid biosynthesis and lipid-associated metabolism. Twelve significant pathways with the BioCyc database were called, including: anandamide degradation, bile acid biosynthesis, glycerol-3-phosphate shuttle, triacylglycerol biosynthesis, CDP (Cytidine diphosphate) diacylglycerol I and II, glycerol degradation, estrogen biosynthesis, cardiolipin biosynthesis, and biosynthesis of serotonin and melatonin (Table 2). Seven significant pathways were identified with the KEGG database including: primary bile acid synthesis, steroid hormone biosynthesis, glycerolipid metabolism, steroid biosynthesis, biosynthesis of unsaturated fatty acids, glycosaminoglycan degradation, and tryptophan metabolism (Table 2). Full pathway lists and details are shown in Appendix A. Of 839 unique KEGG metabolites, 536 of them were statistically significant. The top 30 metabolites identified by correlation analysis and the top 15 biomarkers (class accuracy 100%, AUC = 1) are shown in Appendix A. The peak intensity of these metabolites was subjected for enrichment analysis and pathway analysis on the MetaboAnalyst platform. The *p*-values from the enrichment analysis were adjusted by the Holm–Bonferroni method and by the false discovery rate. The impact value of each pathway was calculated by pathway topology analysis (as stated in the Materials and Methods section). As a result, 73 pathways were picked (Appendix A), and 67 pathways showed statistical significance (FDR < 0.05, False Discovery Rate). The top 25 pathways are shown in Figure 2A and Appendix A. The correlation and network connection of all metabolites and pathways are shown in Appendix A.

#### 3.1.2. BioPersist Enhances the Efficiency of Bile Acid Recycling through Enterohepatic Circulation

Of the 67 significant pathways, three super pathway families were grouped and well noted: steroid metabolism, prostaglandin metabolism, and energy metabolism. Steroid super family pathways include androgen and estrogen metabolism, the steroid biosynthesis pathway, the steroid hormone biosynthesis pathway, and bile acid biosynthesis. Colitic mice treated with BioPersist showed an increased cholesterol level in the colon yet decreased metabolites involved in cholesterol degradation pathways (Appendix A). Cholesterol downstream pathways contained multiple biological functions such as bile acid biosynthesis and steroid hormone biosynthesis. Of the bile acid biosynthesis pathway, more than 17 metabolites were observed (Appendix A). The sub-network projected to the bile acid pathway showed the significant metabolites between the two comparisons (Appendix A), with the green highlighted as statistically significant features. The correlation map indicated that three metabolites were negatively correlated (−1, −0.5) (Appendix A) but not statistically significant (data not shown). Of all the important metabolites of the bile acid biosynthesis pathways, cholic acid, the major primary bile acid, conjugated primary and secondary bile salts including glycochenodeoxycholate, taurochenodeoxycholate, glycocholate, glycolithocholate, and glycolithocholate-3-sulfate, which were all decreased (Figure 2B,C) in colitic mice treated with BioPersist. This illustrated an efficient bile acid recycling in enterohepatic circulation with the intervention of BioPersist. Taurocholate, the conjugated primary bile salt, was increased in mice treated with BioPersist (Figure 2C). Taurine, the precursor of taurocholate, was also increased in the BioPersist group (Figure 2C). Given that taurine is the functional metabolite that positively regulates microbial composition and promotes the production of beneficial short-chain fatty acids [37], increased taurine may be indicative of the benefits that BioPersist elicits during colitis. Similarly, the reduction of bile acid accumulation, including the decreased glycolithocholate and its sulfate conjugate as well as other primary bile acids, suggests BioPersist is beneficial in colitic mice.

#### 3.1.3. BioPersist Attenuates Inflammation in Colitic Mice

The significant features involved in the prostaglandin biosynthesis pathway were shown in its corresponding subnetwork (Appendix A). The correlation between metabolites is presented in Appendix A, yet no statistically significant correlations were annotated (data not shown). To further investigate the change of the intestine-derived metabolites, individual metabolites were called to evaluate the pro-inflammatory response between the two treatments. Mice treated with BioPersist exhibited decreased accumulation of arachidonic acid, the precursor of pro-inflammatory eicosanoids, and its downstream prostaglandin (PG), including PGG_2_, PGH_2_, PGE_2_, PGI_2_, PGD_2_, and thromboxane A_2_ and leukotriene A_4_ (Appendix A, Figure 3). Despite increased PGF_2a_, the consistency of other decreased pro-inflammatory eicosanoids revealed that BioPersist intervention attenuated the inflammatory response in the DSS colitis mouse model. From this data, BioPersist appears to suppress colon-derived metabolites involved in inflammation during colitis.

#### 3.1.4. BioPersist Reduces Stress Hormones and Signals through the Gut–Brain Axis

As demonstrated in Figure 4A, several major steroid hormones including the intermediate products were altered with the treatment of EcN compared to BioPersist when the subnetwork was built referencing C21 steroid hormone biosynthesis and metabolism. Compared to the mice treated with EcN, BioPersist reduced the levels of major stress hormones, including cortisone, corticosterone, androsterone, L-adrenaline, and aldosterone (Figure 4B,C). Taken together, not only major steroid hormones were altered with the treatment of BioPersist but also their intermediate metabolites involved in the same pathway. This revealed that BioPersist exhibited a global effect on the steroid biosynthesis pathway. Aldosterone regulates water and salt retention in the kidney [38], which is upregulated with the increased production of reactive oxygen species and inflammation [39]. Similarly, androsterone is one of the downstream metabolites of testosterone, which is shown to be elevated under stress, such as brain injury [40]. The other three major stress hormones, cortisone [41], corticosterone [42], and adrenaline [43], are well-known to increase blood flow and muscle contraction towards the brain, indicating an amplified and accelerated communication between intestine and brain [44]. On the contrary, anti-stress and anti-depressant hormones and neurotransmitters were increased in mice treated with BioPersist, such as acetylcholine and serotonin (Figure 4D). In addition, the steroid hormone biosynthesis pathway relates to many other upstream and downstream pathways including androgen and estrogen biosynthesis and metabolism, squalene and cholesterol biosynthesis, histamine metabolism, and bile acid metabolism. Hence, these pathways were selected to observe the correlation among the metabolites (Appendix A). The correlation analyses indicated statistically significant correlations between estrone and carnosine, and among urocanate, L-histidine, and 4-imidazolone-5-propanoate (Appendix A). Histamine metabolism, along with the altered hormones, indicated a linear regulatory relationship between intestine and brain. A number of studies have reported the importance of the gut–brain axis, focusing on the interaction of bi-directional communication between the central and the enteric nervous systems [44]. This interaction highlights the mutual regulation of gut microbiota and the cognitive center of the brain through the alteration of hormonal, immune, and neural responses.

Tryptophan metabolism in the colon has been implicated in the gut–brain axis network [45,46]. Mapping the network based on tryptophan metabolism disclosed the change of all metabolites with the treatment of EcN and BioPersist (Figure 4F), with two sets of metabolites showing a positive correlation (Appendix A). However, only 4,6-dehydroxyquinoline and 4,8-dehydroxyquinoline demonstrated statistical significance (Appendix A). When looking into the metabolites involved in tryptophan metabolism, the primary branches involved in this metabolism include kynurenine, indole, and serotonin production (Figure 4E). The kynurenine pathway is the primary catabolic route of tryptophan, which occupies 90% of the tryptophan degradation [47]. However, kynurenine has also been demonstrated to have anti-inflammatory responses by promoting the secretion of IL-10 from Treg cells [48,49]. BioPersist intervention decreased the abundance of kynurenine (Figure 4G). Given that increased kynurenine is detected in patients with ulcerative colitis [50], this may indicate that BioPersist attenuates the disease activity of colitis. In addition, decreased N-acetyl-serotonin and 5-hydroxy-tryptophan (Figure 4G), as well as elevated serotonin (Figure 4D) suggesting an increased production of tryptophan derived serotonin, is implicated in stress reduction and a lower inflamed mucosa caused by DSS [51]. Indole-3-acetate, the downstream product of the indole pathway, has been shown to suppress gut dysbiosis [52]. Increased indole-3-acetate in mice administrated with BioPersist may indicate a positive regulation in restoring and maintaining the homeostasis of microbial composition in the colon of the colitis mouse model.

### 3.2. The Comparison between L. reuteri and BioColoniz

To understand the potential benefits enhanced by BioColoniz during colitis, the colonic metabolites were evaluated from colitic mice treated with BioColoniz and its unmodified parent *L. reuteri*. A total of 97,560 features were selected, with 651 unique metabolites annotated with the BioCyc mouse database and 764 metabolites identified with the KEGG mouse database. Overall, no significant changes were observed between BioColoniz and the parental strain. PCA analysis showed overlap between 95% confidence intervals with only a small proportion of the variance explained by the model, with the first three PCs accounting for 27%, 22.8% and 16.3% of the variance (Appendix A), despite distinct differences with the PLS-DA (Appendix A). Cross validation of the PLS-DA showed the R^2^ as 0.99295, but Q^2^ only had 0.24281. K-means clustering (Appendix A) showed the mixture of two groups in the two major clusters. Cluster 1 had two samples from the BioColoniz group and one sample from the *L. reuteri* group. Cluster 2 included two samples from each of the two groups. Cluster 3 contained only one sample from the *L. reuteri* group. No significant metabolite was identified with a *t*-test analysis (Appendix A). Only three significant metabolites were observed with the differential volcano plot analysis: C03824 (2-aminomuconate semialdehyde), C02918 (1-methylnicotinamide), and C01152 (3-methylhistidine) (Appendix A), whereas no significant pathways were identified by functional analysis. Similarly, no significant features were identified with the EBAM analysis (Appendix A), yet eight features were statistically significant with the SAM analysis (Appendix A). Overall, we did not detect significant differences between the colon-derived metabolites from BioColoniz and its parent strain in the DSS model of colitis.

### 3.3. The Comparison between BioColoniz and BioPersist

#### 3.3.1. The Administration of BioColoniz and BioPersist Shows Discriminate Features from Intestine-Derived Metabolites between Each Other

The comparison of the two LBPs yielded 793 and 672 unique metabolites annotated with the mouse KEGG and the BioCyc databases, respectively. PCA (Figure 5A) and PLS-DA (Figure 5B) analyses with the top three principal components both indicated good separation between the two genetically modified probiotic groups. The PCA PC1 accounted for 35.9% of the variability in the dataset with the unsupervised method, with PC2 and PC3 explaining a further 18% and 13.3%, respectively. In the PLS-DA, Component 1 accounted for 35.8% of the variability, with Components 2 and 3 showing 13.2% and 12.8%, respectively. Cross validation scores for the PLS-DA showed a good model performance with R^2^ and Q^2^ values of 0.999 and 0.874, respectively. The SAM analysis showed 46 significant features (Figure 5C). However, the EBAM analysis indicated zero significant features between the two groups (data not shown). Functional analysis identified four significant pathways: steroid hormone biosynthesis, porphyrin and chlorophyll metabolism, arachidonic acid metabolism, and primary bile acid biosynthesis (Figure 5D, Appendix A).

Enrichment analysis of the pairwise comparison identified 17 significant pathways with the filtered dataset of 793 unique KEGG IDs (Table 3). The monoisotopic metabolites (Appendix A) were used for pathway analysis. With the pathway analysis module in the MetaboAnalyst platform, 79 pathways were annotated, with 56 of them being statistically significant (FDR < 0.05) (Appendix A). The top 25 pathways exported from the MetaboAnalayst (v 5.0) platform are shown in Table 5. The pathways and metabolite networks are shown in Appendix A.

#### 3.3.2. BioPersist Increases the Production of Tryptophan-Derived Serotonin

The data above revealed that BioPersist increased the tryptophan-derived serotonin production compared to EcN (Figure 4D). Similarly, the BioPersist treatment increased the concentration of serotonin and 6-hydroxymelatonin, a downstream metabolite of melatonin, compared to the treatment of BioColoniz (Table 4, Appendix A). The integrated tryptophan pathway showed many distinct features that were altered with the treatment of the two LBPs (Appendix A) such as tryptamine and metabolites involved in indole pathways. Of all correlated metabolites, seven pairs of metabolites including methylserotonin/indole-3-acetaladehyde, indole-3-acetaldehyde/2-amino-3-carboxymuconate-semialdehyde, 3-hydroxyanthranilate/L-kynurenine, indole-3-acetate/5-hydroxyindoleacetaldehyde, 4,5-dihydroxyquinoline/4,6-dihydroxyquinoline, 4(2-amino-3-hydroxyphenyl)-2,4-dioxobutanoate/5-hydroxy-L-tryptophan, and 5-hydroxykynurenamine/3-hydroxykunurenamine were significantly correlated together, with 3-hydroxyanthanilate and L-kynurenine showing a negative correlation (Appendix A). Several other indole compounds were upregulated in mice treated with BioPersist compared to that of BioColoniz (Table 4), which was similar to the pairwise comparison between the EcN and BioPersist groups. Taken together, it appears that BioPersist, unlike BioColoniz, can signal through the colon-derived serotonin pathway.

#### 3.3.3. BioPersist Attenuates Pro-Inflammatory Responses in DSS Treated Mice

Studies have shown that elevated histamine degradation, or anti-histamines, helps to ease gastrointestinal discomfort such as diarrhea and abdominal pain [53]. By comparing the metabolites between two genetically modified probiotics treatments, we observed that metabolites from both treatments had a full coverage of the histamine degradation pathway (Appendix A). Two major downstream histamine metabolites (N-methylhistamine and imidazole-4-acetate) were increased with the treatment of BioPersist (Table 5). Because of the tight connection of inflammatory metabolism, antioxidant, and histamine metabolism pathways, the metabolite correlations among these pathways were calculated and mapped (Appendix A). Six sets of metabolites were statistically significant (Appendix A). Interestingly, the products of alternative arachidonic acid-derived metabolites, 8,9-dihydroxyicosatrienoic acid (DHET), 11,12-DHET, 5,6-DHET, and 14,15-DHET, were all positively correlated with each other. DHET metabolites have been shown to rescue chemotaxis, especially MCP-1 derived chemotaxis [54]. Arachidonic-acid-derived DHETs were all decreased in mice treated with BioPersist. Similarly, trihydroxy-eicosatrienoic acid (THETA), the metabolite produced by 15-lipoxygenase, was positively correlated with the prostaglandin metabolites (Appendix A). These data reveal that BioPersist attenuates the inflammatory response. In support of this, mice treated with BioPersist significantly increased the production of spermine, a potent antioxidant and anti-inflammatory agent (Table 5). The well-known glutathione also displayed an increasing trend (*p* = 0.061) in mice treated with BioPersist. Therefore, with the reduced pro-inflammatory metabolites and inducted antioxidants, BioPersist may shed light on attenuating the inflammatory response due to the treatment of DSS.

#### 3.3.4. Mice Treated with BioPersist Reduce Primary Bile Acid Accumulation in the Colon

We have already shown that BioPersist had increased bile acid recycling efficiency with low accumulation of primary bile acids in the colon compared to that of EcN. Here, BioPersist treatment again showed the similar alteration when compared to the mice treated with BioColoniz (Appendix A). With the same concentration of cholesterol between the two LBPs, the production of major bile acids was reduced, including chenodeoxycholate, taurochenodeoxycholate, taurocholate (*p =* 0.053), and glycocholate in the BioPersist group (Table 6). However, glycochenodeoxycholate was increased in mice treated with BioPersist (Table 6). The intermediate metabolites of producing bile acids were all downregulated. Therefore, with the same concentration of the precursor of bile acid, the decreased accumulation of bile acid derivatives in the colon may represent better bile acid recycling through enterohepatic circulation.

### 3.4. BioPersist Initiates the Process of Tissue Repair via Increased Cell Proliferation

The DSS-induced colitis mice model damages the colon epithelium and develops severe inflammation, particularly in the distal colon [55]. The severe inflammatory response impairs the integrity of the protective mucus and suppresses tissue regeneration, resulting in an exaggerated loss of epithelium integrity. In our study, mice treated with BioPersist showed increased purine and pyrimidine metabolism, with elevated production of adenine (A), guanosine (G), uracil (U), and thymine (T) (Table 7). Two sub-networks of purines (Figure 6A) and pyrimidines (Figure 6B) also demonstrated major changes in A, G, U, T, and cytidine as well as other major intermediate metabolites. In particular, the correlation analysis (Figure 6C) disclosed similar results to the individual metabolites’ expression. As an example, adenine and CTP (Cytidine-5’-triphosphate), dCMP (Deoxycytidine monophosphate) and cytidine, and deoxycystidine and cytidine were positively correlated, with both sets of metabolites upregulated in mice treated with BioPersist. In addition, biosynthesis of O-glycans also helps to restore and maintain the mucus barrier function [56]. Tn antigen, as well as (Gal)1 (GalNAc)1 (Neu5Ac)2 (Ser/Thr)1, were increased in mice treated with BioPersist (Table 7). With increased free nucleic acids, as well as their upregulated biosynthesis pathway, this could suggest that BioPersist helps mice to restore intestinal tissues damaged during the DSS challenge.

## 4. Discussion

LBPs comprise a growing segment of the IBD market [57]. Single bacteria or consortiums of commensal bacteria are considered safe but they lack consistent clinical efficacy for IBD [3], perhaps because of their inability to persist in the highly inflamed IBD intestinal environment [2,58,59]. In this study, we enhanced the persistence and colonization of EcN and *L. reuteri*, to evaluate their effect on modulating colon-derived metabolites. While we did not identify metabolic pathways specific to BioColoniz, we were able to generate several hypotheses about the mechanism in which BioPersist (compared to EcN and BioColoniz individually) may provide therapeutic potential for the treatment of IBD. The hypotheses include that, during colitis, BioPerist:Attenuates the inflammatory response by suppressing the production of pro-inflammatory eicosanoids;Provides beneficial effects to the intestine by increasing the production of antioxidants;Improves the stress response through the reduction of stress hormones and elevation of anti-depressant hormones via the gut–brain axis;Restores epithelial and mucosal tissue damage with increased purine and pyrimidine metabolism;Enhances the efficiency of bile acid recycling through enterohepatic circulation.

EcN is one of the most widely used probiotics for the treatment of intestinal diseases such as irritable bowel syndrome [60] and ulcerative colitis [61]. The mechanisms through which EcN provides benefits to various intestinal conditions are diverse. Reports have shown that EcN enhances the bioactivity of the tryptophan-derived serotonin pathway [62] and suppresses the enzymes involved in the production of pro-inflammatory eicosanoids [63]. In our study, the enhanced persistence of EcN (BioPersist) further consolidated these benefits, with increased serotonin and decreased proinflammatory prostaglandins.

Here, we found that BioPersist intervention in colitic mice resulted in suppressed metabolites involved in the pro-inflammatory eicosanoid pathway. The high concentration of proinflammatory eicosanoids, as well as their converting enzymes (COX1 and COX2), have been observed in a number of rodent colitis models and clinical subjects [64,65,66,67]. During colitis, inflammatory prostaglandins exacerbate intestinal inflammation and induce the dysregulation of Treg cells known to suppress the damaging inflammatory response in the intestine [68]. Controlling inflammation is always the first approach for treatment of IBD. The most historical and classic anti-inflammatory drugs, NSAIDs nonsteroidal anti-inflammatory drugs (NSAIDs), however, are not recommended for patients with IBD [69,70]. NSAIDs reduce inflammation through blockage of COX2 [71]. However, their well-known side effect of damaging the intestinal epithelial layer cannot be neglected. With the constructive enzyme COX1 and the inducible enzyme COX2, arachidonic acid is metabolized to proinflammatory eicosanoids including series II prostaglandin, leukotrienes, and thromboxane [71]. One of the well-known mechanisms of the side effects is to completely block the formation of pro-inflammatory eicosanoids via COX2 inhibition [69,70]. Although the series II prostaglandins are pro-inflammatory, they are also essential to maintain the integrity and defense of the epithelial and mucosal layer [72]. Introducing NSAIDs to a DSS-induced colitis mice model has shown to exaggerate the manifestations of colitis [73]. Therefore, maintaining low levels, instead of complete deletion, of prostaglandins is essential to protect the integrity and the functionality of the intestine. BioPersist reduces prostaglandin production from their precursor, arachidonic acid, to its downstream products. Yet, BioPersist did not block prostaglandins like NSAIDs; instead, it achieved the goal of maintaining the prostaglandins at a biologically relevant low level and, at the same time, limited the prostaglandin-derived inflammatory response.

BioPersist, unlike BioColoniz, activates antioxidants in the DSS murine model of colitis. Inflammation and oxidative stress cause and perpetuate tissue damage in any inflammatory diseases. Inflammation triggers the oxidative stress, which in turn exaggerates the inflammatory response. During colitis, there are excessive free radicals such as reactive oxygen species and reactive nitrogen species, which further deplete the defensive potential granted by antioxidants. Oxidative stress can cause further cellular damage including lipid peroxidation, DNA damage, and protein denaturation [74], which may further develop inflammatory metabolic diseases such as IBD and cancer [75,76]. Hence, antioxidant administration has been considered a potential strategy to scavenge free radicals, effectively acting like an anti-inflammation agent. Clinical reports have indicated deficient extracellular antioxidants levels in the blood and colons of patients with IBD compared to healthy people [77]. Glutathione (GSH) is a well-known potent antioxidant, which has been widely studied in several inflammatory states including IBD [75,78,79]. Supplementation of GSH has shown to reduce disease activity in TNBS-induced colitic rats, improve mucosal function [80], and inhibit the production of malondialdehyde, a lipid peroxidation product [81]. Spermine is another potent antioxidant that helps to restore the concentration of endogenous antioxidant glutathione. A rodent study showed that spermine enhanced the production of GSH, catalase, and anti-hydroxyl radical by 1.32%, 38.68%, and 15.53%, respectively [82]. Spermine supplementation in piglets induced beneficial changes to the gut microbiome through the increased production of GSH in the ileum [83]. BioPersist may act as an effective tool to restore the loss of antioxidants caused by the inflamed intestine during colitis.

Intestine-derived serotonin is a hormone produced by enterochromaffin cells regulated by the gut microbiome, which plays a role in regulating mood and GI motility [84]. Reduced levels of the anti-depressant hormones and their receptors can result in depression and anxiety [85]. The first-line clinical practice for depression has long been prescribed selective serotonin reuptake inhibitors (SSRIs), which block the reabsorption of serotonin [86,87]. Other alternative therapeutic solutions include introducing serotoninergic postsynaptic receptors [88] and inhibition of the serotonin degradation with monoamine oxidase inhibitors [88,89]. The commonality of all these strategies is to elevate the serotonin concentration in serum, resulting in a positive effect on mood. Serotonin biosynthesis and secretion are detected in both the central nervous system and the peripheral nervous system, with 95% of serotonin produced in the intestine through the local microbiome [90]. Tryptophan is the only precursor of serotonin produced in the intestine [91]. The shortage of the microbial-derived enzymes converting tryptophan to serotonin leads to decreased GI motility including colonic peristaltic reflex [92], which is one classic colonic disruption in colitis patients [93]. Additionally, patients with IBD have a higher risk of developing psychiatric comorbidities including depression and dementia through the gut–brain axis [94]. Despite the limited evidence on the correlation between the modulation of serotonin levels and IBD occurrence, a clinical trial in Denmark demonstrated beneficial effects of serotonin administration on improving the IBD relapse rate [95]. In addition, people with depression and IBD both showed increased production of stress hormones such as cortisol, which may contribute to chronic inflammation [96,97]. In this context, BioPersist intervention may help reduce stress hormones and the inflammatory response, promoting colitis remission unlike the commercially available EcN. Hence, we hypothesize that enhanced persistence of EcN (with BioPersist) helps to ameliorate colitic-induced inflammation via the gut–brain axis.

Endogenous reactive oxygen and nitrogen species must be kept in homeostasis to avoid further tissue damage, especially DNA damage. Lack of DNA repair exaggerates the pre-existing inflammatory response, and further progression would develop into colonic cancer. A defect in DNA repair mechanism worsens DSS-induced ulcerative colitis and results in larger colonic tumors [98]. Mice lacking OGG1 (8-oxoguanine DNA glycosylase) increased the incidence of carcinogenesis with the treatment of DSS [98]. DNA repair potential is positively correlated with the GC content of the genome [99]. Here, we found that BioPersist increased guanosine and cystine concentration, suggesting its ability to restore DNA damage due to the DSS challenge. In addition, the close correlations among the major metabolites of purine and pyrimidine metabolisms integrated the upregulated de novo biosynthesis and alternative salvage pathways. As an example, adenosine and dAMP showed a positive correlation, which connects the salvage pathway and the de novo biosynthesis of purine metabolism. In addition, purine and pyrimidine metabolism are synchronized, with significant positive correlations between adenine and CTP, adenosine and CTP, UTP and dADP, and 5,6-dihydrouracil and inosine monophosphate (IMP). The increased available nucleotides and their intermediate metabolites may reflect the elevated DNA and tissue repair system; however, further validation experiments are required to consolidate this hypothesis.

Bile acid malabsorption or decreased bile acid recycling is often neglected in patients with IBD, yet many observations have been established in different rodent models and clinical human studies. Active IBD patients have increased conjugated bile acids accumulating in feces due to the decreased microbial richness in the distal ileum and colon [100,101]. Similarly, many colitis rodent models present the same observations. In the DSS-induced colitis model, the accumulation of cholic acid in feces has been reported [102]. TNBS-induced colitis in rats has also shown increased bile acid accumulation in feces due to decreased expression of bile acid transporters, resulting in the suppression of bile acid recycling [103]. Similarly, supplementation of deoxycholic acid exacerbates the inflammatory response in DSS-treated mice [104]. In clinical studies, enrichment of primary bile acid cholate and conjugates are detected in dysbiotic samples [105]. Bile acid recycling defect is also shown in patients with diarrhea [106]. Hence, increased bile acid excretion or the reduced bile acid recycling are the concomitant signs of IBD. In this study, BioPersist intervention resulted in decreased accumulation of bile acids, which may indicate that BioPersist promotes efficient bile acid recycling via enterohepatic circulation, reduced bile acid synthesis through cholesterol, or improved homeostasis of intestinal microbial growth. The promising metabolism of bile acid reflects the important role of BioPersist on modulating the gut–liver axis through enterohepatic circulation, which is an essential bridge to ensure multi-organ drug metabolism, hormone and energy metabolic homeostasis, and detoxification. However, validation of these results and its definitive mechanism requires further investigation.

## 5. Conclusions

In summary, discriminate pathways were annotated with two pairwise comparisons: EcN vs. BioPersist and BioPersist vs. BioColoniz. Two universal benefits stand out with the treatment of BioPersist from the two different comparisons. BioPersist showed distinct benefits on (1) recycling bile acid through enterohepatic circulation and (2) attenuating the inflammatory response due to the administration of DSS. Besides, the enhanced persistence of EcN in mice reduced stress and activated the production of anti-stress hormones and may explicate a beneficial effect on tissue restoration compared to the mice treated with EcN. When comparing to BioColoniz, BioPersist may inhibit the inflammation-derived DNA damage by activating purine and pyrimidine metabolism. Therefore, BioPersist can not only enhance the pre-existing benefits inherited from its parental strain EcN but can also provide an additional promising effect on improving colitis by restoring inflamed tissue or by modulating the gut–brain axis.

## Figures and Tables

**Figure 1 biomolecules-11-00738-f001:**
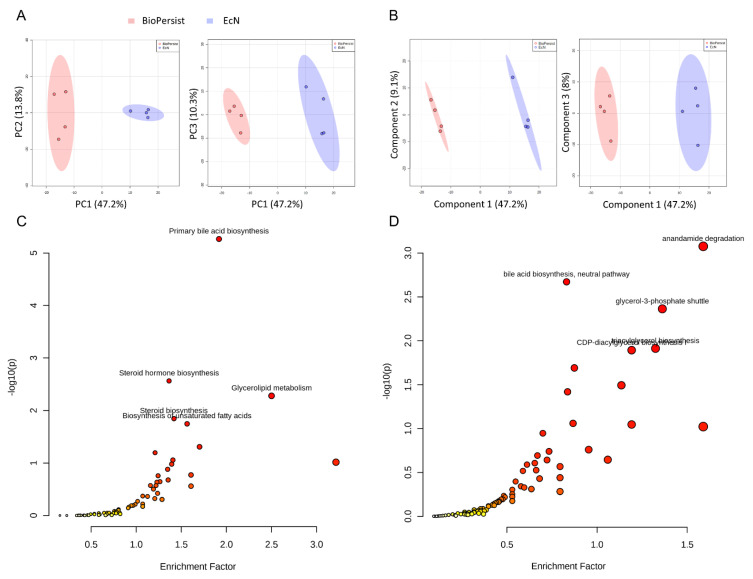
Signature pathways of mice treated with EcN and BioPersist. (**A**) PCA score plots with PC1 against PC2 and PC1 against PC3 for mice treated with EcN and BioPersist. (**B**) PLS-DA score plots with Component 1 against Component 2 and Component 1 against Component 3 for mice treated with EcN and BioPersist. (**C**) Significant pathways annotated with mouse KEGG and (**D**) BioCyc. *n* = 4 for each group.

**Figure 2 biomolecules-11-00738-f002:**
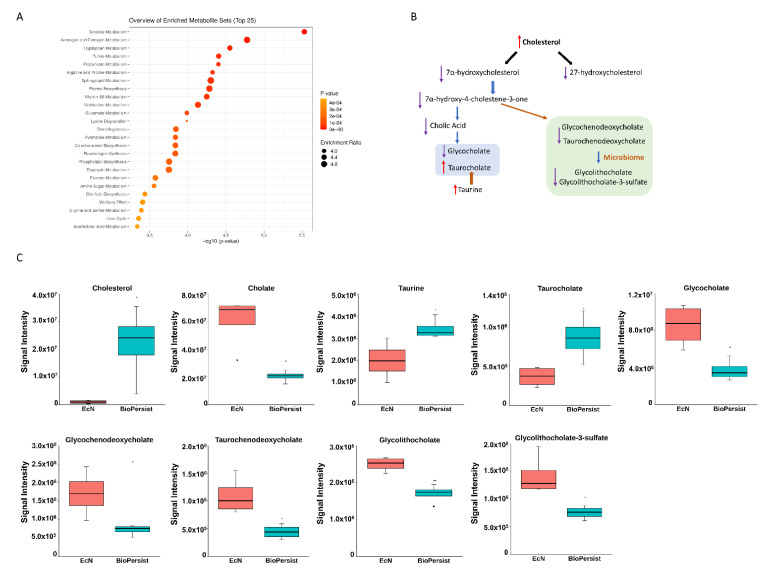
BioPersist decreases the accumulation of bile acids in the colon. (**A**) The top 25 pathways calculated with pathway analysis on MetaboAnalyst. (**B**) Summary of bile acid biosynthesis. Conjugated bile acids are highlighted in blue and secondary bile acids in green. (**C**) Individual metabolites involved in the bile acid biosynthesis pathway. *n* = 4 per group. * *p* < 0.05.

**Figure 3 biomolecules-11-00738-f003:**
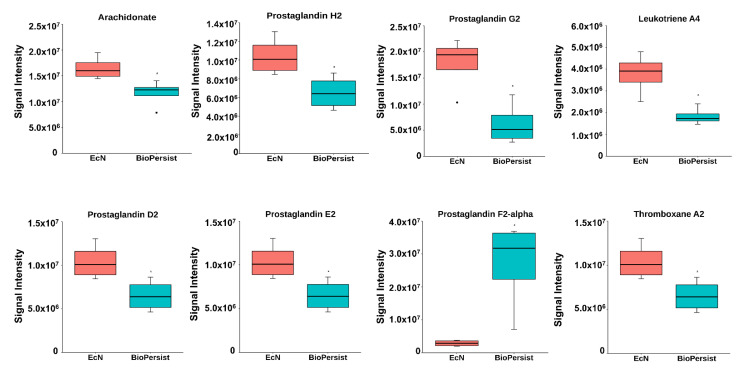
BioPersist reduces the production of pro-inflammatory prostaglandins. Individual metabolites involved in the prostaglandin biosynthesis pathway. *n* = 4 per group. * *p* < 0.05.

**Figure 4 biomolecules-11-00738-f004:**
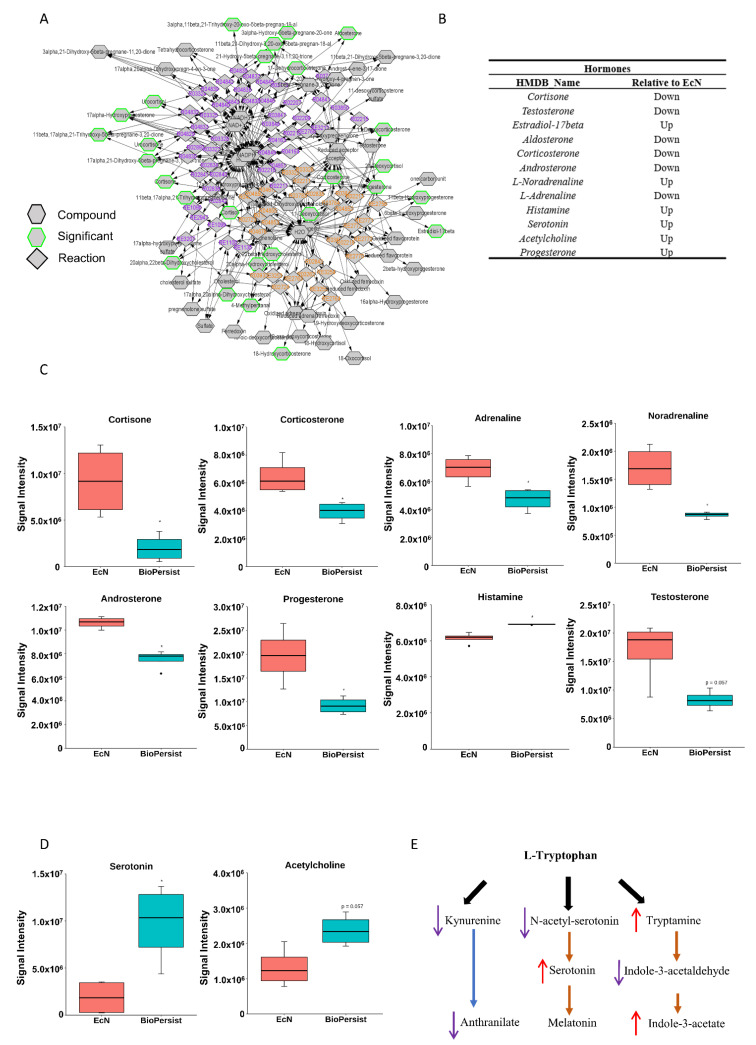
BioPersist decreases production or stress hormones and increases tryptophan derived anti-stress hormones. (**A**) Subnetwork of C21-steroid hormone biosynthesis and metabolism pathway. Metabolites highlighted in green indicated statistically significant features. (**B**) Summary of differential expressions of hormones between mice treated with EcN and BioPersist. (**C**) Concentrations of major stress hormones including cortisone, corticosterone, noradrenaline, adrenaline, aldosterone, testosterone, androsterone, and histamine in mice treated with EcN and BioPersist. (**D**) Serotonin and acetylcholine concentration in mice treated with EcN and BioPersist. (**E**) Schematic summary of three major tryptophan-derived pathways. (**F**) Subnetwork of the tryptophan metabolism pathway. Metabolites highlighted in green indicated statistically significant features. (**G**) Individual metabolites involved in tryptophan degradation pathways. *n* = 4 per group. * *p* < 0.05.

**Figure 5 biomolecules-11-00738-f005:**
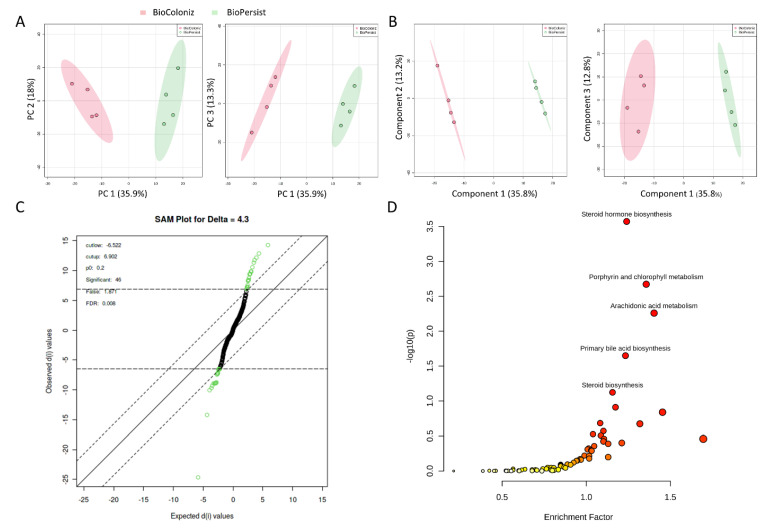
Signature pathways in mice treated with BioColoniz and BioPersist. (**A**) PCA score plots with PC1 against PC2 and PC1 against PC3 for mice treated with BioColoniz and BioPersist. (**B**) PLS-DA score plots with PC1 against PC2 and PC1 against PC3 for mice treated with BioColoniz and BioPersist. (**C**) Significant pathways annotated with the mouse KEGG database. *n* = 4 per group. (**D**) Significance analysis of microarray (SAM) analysis identified 46 significant features of the two probiotics.

**Figure 6 biomolecules-11-00738-f006:**
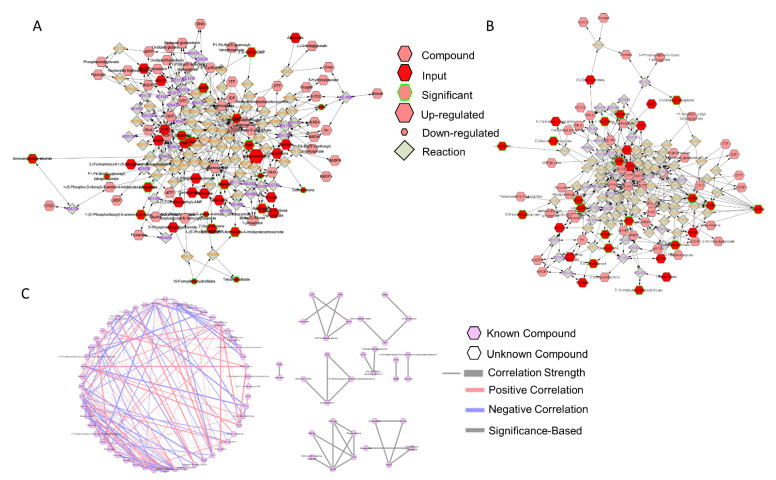
The network of purine and pyrimidine metabolisms. Subnetworks of the (**A**) purine and (**B**) pyrimidine metabolism pathways. Metabolites highlighted in green indicated statistically significant features. (**C**) Correlation of metabolites annotated in the purine and pyrimidine metabolism pathways. Connections in red and blue indicated positive and negative correlation between the two metabolites, respectively. Statistically significant correlations of metabolites were highlighted with grey. *p* < 0.05.

**Table 1 biomolecules-11-00738-t001:** Summary of metabolites hit by the two databases.

KEGG Database		BioCyc Database	
Total hits	Common hits	Total hits	Common hits
839	533	686	483

**Table 2 biomolecules-11-00738-t002:** Significant pathways hit by the mouse BioCyc database.

BioCyc_Pathway	Pathway Total	Hits Total	Hits Sig	*p*-Value
Anandamide degradation	6	6	6	<0.001
Bile acid biosynthesis, neutral pathway	44	44	23	0.002
Glycerol-3-phosphate shuttle	7	7	6	0.004
Triacylglycerol biosynthesis	6	6	5	0.012
CDP-diacylglycerol biosynthesis I	8	8	6	0.012
CDP-diacylglycerol biosynthesis II	8	8	6	0.012
Glycerol degradation IV	8	8	6	0.012
Biosynthesis of estrogens	20	20	11	0.020
Cardiolipin biosynthesis II	7	7	5	0.032
Cardiolipin biosynthesis I	7	7	5	0.032
Biosynthesis of serotonin and melatonin	19	19	10	0.038
Serotonin and melatonin biosynthesis	19	19	10	0.038
**KEGG_Pathway**	**Pathway Total**	**Hits Total**	**Hits Sig**	***p*-Value**
Primary bile acid biosynthesis	57	57	34	<0.001
Steroid hormone biosynthesis	137	137	60	<0.001
Glycerolipid metabolism	9	9	7	0.005
Steroid biosynthesis	68	68	30	0.011
Biosynthesis of unsaturated fatty acids	37	37	18	0.016
Glycosaminoglycan degradation	17	17	9	0.045
Tryptophan metabolism	125	125	47	0.051

**Table 3 biomolecules-11-00738-t003:** Significant pathways hit with 793 unique KEGG IDs.

KEGG Pathway	Total	Hits	Raw *p*	FDR
Steroid hormone biosynthesis	85	70	<0.001	<0.001
Steroid biosynthesis	42	38	<0.001	<0.001
Arachidonic acid metabolism	36	32	<0.001	<0.001
Tryptophan metabolism	41	34	<0.001	<0.001
Tyrosine metabolism	42	32	<0.001	0.007
Arginine and proline metabolism	38	29	<0.001	0.010
Histidine metabolism	16	14	0.002	0.024
Galactose metabolism	27	21	0.003	0.028
Caffeine metabolism	10	9	0.010	0.088
Phenylalanine metabolism	10	9	0.010	0.088
D-glutamine and D-glutamate metabolism	6	6	0.015	0.117
One carbon pool by folate	9	8	0.019	0.123
Vitamin B6 metabolism	9	8	0.019	0.123
Retinol metabolism	17	13	0.024	0.142
Metabolism of xenobiotics by cytochrome P450	68	42	0.030	0.169
Valine, leucine, and isoleucine biosynthesis	8	7	0.035	0.181
Pentose and glucuronate interconversions	18	13	0.047	0.230

**Table 4 biomolecules-11-00738-t004:** Significant features involved in the tryptophan metabolism pathway.

Common Name	KEGG ID	Significance	Relative to BioColoniz
4,6-dihydroxyquinoline	C05639	0.02	Down
4-(2-aminophenyl)-2,4-dioxobutanoate	C01252	0.0048	Down
4,8-dihydroxyquinoline	C05637	0.02	Down
Tryptamine	C00398	0.0382	Down
Indolepyruvate	C00331	0.0325	Up
Indole-3-acetaldehyde	C00637	0.037	Up
5-hydroxyindoleacetate	C05635	0.0012	Down
Formylanthranilate	C05653	<0.001	Up

**Table 5 biomolecules-11-00738-t005:** Significant features involved in histamine degradation and glutathione metabolism pathways.

Common Name	KEGG ID	Significance	Relative to BioColoniz
L-histidine	C00135	0.0023	Down
Histamine	C00388	0.057	Up
N-methylhistamine	C05127	0.0276	Up
Imidazole-4-acetate	C02835	0.0336	Up
Methylimidazole	C05827	<0.001	Up
spermidine	C00315	0.0173	Up
Spermine	C00750	0.0012	Up
Cyc-gly	C01419	0.0219	Down
L-cysteine	C00097	0.0058	Down
Glutathione	C00051	0.0061	Up

**Table 6 biomolecules-11-00738-t006:** Significant features involved in the primary bile acid biosynthesis pathway.

Common Name	KEGG ID	Significance	Relative to BioColoniz
25-hydroxycholesterol	C15519	<0.001	Down
4-cholesten-7alpha,12alpha-diol-3-one	C17339	0.0011	Down
7alpha-hydroxycholest-4-en-3-one	C05455	0.036	Down
Chenodeoxycholate	C02528	<0.001	Down
3alpha,7alpha,12alpha-trihydroxy-5beta-cholestan-26-al	C01301	0.0584	Up
7alpha-hydroxycholesterol	C03594	<0.001	Down
Cerebrosterol	C13550	<0.001	Down
3alpha,7alpha-dihydroxy-5beta-cholestanate	C04554	0.0584	Up
3alpha,7alpha,12alpha-trihydroxy-5beta-cholestanoate	C04722	0.0513	Down
3beta-hydroxy-5-cholestenoate	C17333	0.0011	Down
Glycochenodeoxycholate	C05466	0.0062	Up
Taurochenodeoxycholate	C05465	0.0018	Down
7alpha,26-dihydroxy-4-cholesten-3-one	C17336	0.0011	Down
7alpha,25-dihydroxy-4-cholesten-3-one	C17332	0.0011	Down
Glycocholate	C01921	0.0361	Down
Taurocholate	C05122	0.053	Down
7alpha,24-dihydroxy-4-cholesten-3-one	C17331	0.0011	Down

**Table 7 biomolecules-11-00738-t007:** Significant features involved in purine, pyrimidine, and O-glycan biosynthesis pathways.

Common Name	KEGG ID	*p*-Value	Relative to BioColoniz
alpha-D-ribose 1-phosphate	C00620	0.0152	Down
5-phosphoribosylamine	C03090	<0.001	Up
2-(formamido)-N1-(5′-phosphoribosyl)acetamidine	C04640	0.0012	Up
1-(5′-phosphoribosyl)-5-amino-4-(N-succinocarboxamide)-imidazole	C04832	0.0017	Up
N6-(1,2-dicarboxyethyl)AMP	C03794	<0.001	Up
Inosine	C00294	0.0139	Up
Hypoxanthine	C00262	0.0604	Up
dAMP	C00360	0.0074	Up
dADP	C00206	0.003	Up
Adenine	C00147	<0.001	Up
Guanosine	C00387	<0.001	Up
Guanine	C00242	<0.001	Down
Xanthine	C00385	0.006	Up
Xanthosine	C01762	0.0111	Up
dGTP	C00286	0.01	Up
UTP	C00075	0.0044	Down
Uridine	C00299	0.0258	Up
Deoxyuridine	C00526	<0.001	Down
Uracil	C00106	0.0013	Up
5,6-dihydrouracil	C00429	0.0374	Down
3-ureidopropionate	C02642	<0.001	Down
Thymine	C00178	0.0336	Up
(R)-3-amino-2-methylpropanoate	C01205	0.0379	Up
Tn antigen	G00023	0.0031	Up
T antigen	G00024	0.0135	Down
(Gal)1 (GalNAc)1 (Neu5Ac)2 (Ser/Thr)1	G00027	0.0018	Up

## Data Availability

Data sharing not applicable.

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
