# Peer review of "Metabolomics-Guided Hypothesis Generation for Mechanisms of Intestinal Protection by Live Biotherapeutic Products"

_biomolecules, 2021, doi:10.3390/biom11050738_

Round 1

Reviewer 1 Report

The manuscript entitled “Metabolomics-guided hypothesis….. live biotherapeutic products” by Ye et al, evaluated the beneficial effect of BioPersist an engineered E. coli strain commonly used as a probiotic. Authors claim that the engineered strain is capable of persisting more in the gut and can overcome the adverse effect of an inflamed environment. Due to its persistence, probiotic bacteria are able to regulate several metabolites that regulate key biological pathways that are beneficial to the host.  This is a very interesting study that explored the untargeted metabolomics data and provides several hypotheses for the mechanism of action of the mentioned LBP. This study opens up new questions and perspectives in the field of LBP addressing the mechanism of action of LBP is a must in the future. The manuscript is written very well and in detail. I recommend acceptance of this manuscript after a minor revision.

I have the following questions and suggestions

  1. The current manuscript starts with the metabolomic data, but data to display that BioPersist LBP is really persistent in the colitis model. How long this engineered bacterial can persist in the gut and how it differs from its parent strain in attaching to the gut. Need some information on these to appreciate the manuscript.
  2. Data showing the bacterial count or histopathology of the gut might support the claims made by the authors regarding the persistence. If there is published material from the authors in relation to these please cite the published materials.
  3. Most of the figures especially the network graphs are unclear and do not make any meaning. These figures can be in the supplement file if they have to be presented in a clear format.
  4. The tables can also be moved to supplement as they are representing the same data of signal intensity figures.
  5. There are few typos in the manuscript please go through them to edit carefully. Example Line 526: Identify

Author Response

Thank you for the valuable comments. We have made changes and clarified the questions as below:

1. The current manuscript starts with the metabolomic data, but data to display that BioPersist LBP is really persistent in the colitis model. How long this engineered bacterial can persist in the gut and how it differs from its parent strain in attaching to the gut. Need some information on these to appreciate the manuscript.

Response: The data supporting persistence and its effect in the gut is in the patent application filed.  The data will be expanded in another manuscript on the effect in the gut and its persistence and will not be presented here given its patent protection and public availability in this application (PCT/CA2018/050188). This manuscript focuses on its role on metabolites.

2. Data showing the bacterial count or histopathology of the gut might support the claims made by the authors regarding the persistence. If there is published material from the authors in relation to these please cite the published materials.

Response: Similarly, the data supporting its role in gut models of colitis are in the patent file (PCT/CA2018/050188) and another manuscript describing this is in another manuscript in preparation for publication.  Here we focused on the role of the live biotherapeutics on metabolites to generate hypothesis about its mechanism of action compared to the unmodified parental strains.

3. Most of the figures especially the network graphs are unclear and do not make any meaning. These figures can be in the supplement file if they have to be presented in a clear format.

Response: The network graphs have now been moved to the supplement files and altered in a clear format. Here is a breakdown changes of figures:

1). Figure 2B and Figure 2C (network figures) have been moved to supplementary Figure S5.

2). Figure 3A-3C have been moved to supplementary Figure S6.

3). Figure 4A and 4F stayed since we’d like to show the significant changes of intermediate metabolites of the corresponding pathways. Those metabolites were not presented in box plots. We added the explanation of the corresponding network graphs in LINE 353-355.

4). Figure 6 has been moved to supplementary Figure S11.

5). Figure 7 was kept since the linear correlation between metabolites are important to address the conclusion ‘BioPersist initiated the process of tissue repair via increased cell proliferation’

4. The tables can also be moved to supplement as they are representing the same data of signal intensity figures.

Response: Table 3 has been moved to supplementary Table S4.

Table 4a has been moved to supplementary Table S6.

The rest of tables were kept since there were no corresponding graphs in the main manuscript.

5. There are few typos in the manuscript please go through them to edit carefully. Example Line 526: Identify

Response: We have edited the manuscript carefully.

Reviewer 2 Report

The manuscript describes the changes in vivo gut-metabolite production by two engineered probiotic strains compared to their unmodified parent strains. Gut-derived metabolites were analyzed by UHPLC-MS in colitic mice colon. The data obtained was subject to an extensive in silico analysis.

I recommend publication after the clarification of the following issues:

Although the two engineered strains are patented some details in what kind of changes were made or introduced could be given.

Some details on the live bio-therapeutic preparation should be given. How were they grow and prepared to be fed to the mice?

Please explain why different dosages of live biotherapeutics and protocol (1 x 2.10^9 vs 3 x 3x10^12) were used.

The patent should be listed in the references list.

Author Response

We appreciate the valuable suggestions and comments. We have made changes or clarified the questions as below:

1. Although the two engineered strains are patented some details in what kind of changes were made or introduced could be given.\

Response: These details are in the patent application (PCT/CA2018/050188) which is publicly available.

2. Some details on the live bio-therapeutic preparation should be given. How were they grow and prepared to be fed to the mice?

Response: Thank you for the suggestion. We have added the section of bacterial culture as 2.1 from LINE 98-110.

 ‘EcN and BioPersist were grown in Luria-Bertani overnight (LB, 10 g tryptone, 5 g yeast extract and 10 g sodium chloride dissolved in 1 L of water, adjusting pH to 7.5) for 16 h at 37°C under 180rpm agitation. L. reuteri and BioColoniz were grown in De Man, Rogosa and Sharpe broth (MRS, 10 g peptone, 10 g beef extract, 5 g yeast extract, 20 g dex-trose, 5 g sodium acetate, 1 g polysorbate 80, 2 g dipotassium phosphate, 2 g ammonium citrate, 0.1 g magnesium sulfate, and 0.05 g manganese sulfate dissolved in 1 L of water, adjusting pH to 6.5) for 24 h at 37°C static under anaerobic conditions given by the BD GasPak system (BD Biosciences). The mice were gavaged immediately with the fresh pro-biotics kept at room temperature. Parallel and to confirm the dose given to mice, 0.1ml of the probiotics were plated on 100 x 15 mm agar plates of the corresponding medium and grown for 24 h at 37°C for EcN and BioPersist, and for 48 h at 37°C under anaerobic con-ditions for L. reuteri and BioColoniz.

3. Please explain why different dosages of live biotherapeutics and protocol (1 x 2.10^9 vs 3 x 3x10^12) were used.

Response: This is the growth of each LBP after 24 hr of growth.

4. The patent should be listed in the references list.

Response: We have now listed the application number in the references.

Thank you for all your kind and important comments.

Reviewer 3 Report

I`am fully convinced, that presented manuscript should be published in Biomolecules. Authors presented in very extensive and precise way results of very interesting experimental study on new formulas of probiotics.  Novel aspects are presented in the light of possible clinical application in inflammatory diseases. One remark, I would recommend to remove the last sentence of conclusions as is not directly related to the results of the project.

Author Response

Thank you so much for your comments. We have removed the last sentence from the main manuscript. Please see our revised manuscript.